# Does Entrepreneurship Education Deliver? A Review of Entrepreneurship Education University Programmes in the UK

Matthew Rogers-Draycott [1,*], David Bozward [2], Kelly Smith [3], Mokuba Mave [4], Vic Curtis [5] and Dean Maragh [6]

1 College of Business, Digital Transformation & Entrepreneurship, Birmingham City Business School, Birmingham City University, 4 Cardigan St, Birmingham B4 7RJ, UK

2 Global Banking School, Norfolk House, 84-86 Smallbrook Queensway, Birmingham B5 4EG, UK; dbozward@globalbanking.ac.uk

3 Department of Management, Birmingham Business School, University of Birmingham, Edgbaston, Birmingham B15 2TT, UK; k.j.smith@bham.ac.uk

4 Leicester Castle Business School, De Montfort University, Gateway House, Leicester LE1 9BH, UK; mokuba.mave@dmu.ac.uk

5 College of Business, Law and Social Sciences, University of Derby, Kedleston Rd., Derby DE22 1GB, UK; v.curtis@derby.ac.uk

6 Department of Management, Sheffield Business School, Sheffield Hallam University, Howard St., Sheffield City Centre, Sheffield S1 1WB, UK; d.maragh@shu.ac.uk

* Correspondence: matthew.draycott@gmail.com

**Abstract:** The student is a consumer of education and is motivated by their graduate outcomes. Entrepreneurship provides the opportunity for substantially greater graduate outcomes, but does it deliver? This paper reviews the undergraduate bachelor Entrepreneurship Education Programmes (EEPs) offered by universities in the UK. It explores the active and engaged approaches to learning through the module themes offered and considers the challenges of using routinely collected data to understand the impact of these programmes. By using data which is publicly available, we build a national viewpoint on the subjects that lead to greater continuation, student satisfaction and earning potential. The results of this study provide five key insights about EEPs. First, they focus mainly on entrepreneurship but lack a high proportion of entrepreneurship-specific modules. Comparative analysis with other disciplines is needed for context. Second, the number of entrepreneurship modules generally increases throughout the programme, but researchers face challenges such as ambiguous module naming. Third, EEP students show lower satisfaction than those studying for other business degrees, indicating a potential impact of unique pedagogies. Fourth, higher EEP continuation rates are not evident, although this may be mitigated by more selective entry requirements. Finally, EEP graduates have higher employability rates than their business degree counterparts but lower initial earnings, reflecting their entrepreneurial career paths. From this work, we identified a range of calls for further research and suggestions for practice.

**Keywords:** entrepreneurship education; curriculum design; entrepreneurship pedagogy; student satisfaction; student outcomes; student continuation

## 1. Introduction

The development of Entrepreneurship Education (EE) within universities over the last forty years has been unprecedented, with sustained growth in the number of dedicated chairs, faculty members, undergraduate and postgraduate programmes, conferences, and research scholarships [1–3].

This has been driven by the fact that entrepreneurship has, for some time, been seen as an engine for innovation [4], regional economic growth [5], and job creation [6]. Yet, for all its promise, there is surprisingly little literature that can speak to these impacts [7,8] or connect them to interventions in a meaningful way outside of a small number of specific

contexts [9]. This continues to be the case, even though there have been numerous calls to address this agenda over a prolonged period [10–12]. As [7] concludes, impact research predominantly focuses on short-term and subjective outcome measures.

To understand, at any scale, the impact of EE, we previously suggested that researchers need to add to the numerous, small-scale, mainly qualitative publications that dominate the current discourse with broader studies that explore larger datasets [5].

In this paper, we explored EE programmes (EEPs) in the UK through the use of several publicly available sources, including data scraping of UK university websites for programme information, alongside several national datasets that explore student attitudes to their courses and their employment following graduation.

The aim of this research is to review UK undergraduate EEP provision to understand their impact, with a focus on student continuation, satisfaction, and employability. This aim is achieved by first exploring the literature to generate several hypotheses, which are then tested against the collated datasets. We contend that this original work makes a significant contribution to the literature on the impact of EEPs, especially in a UK context, as these data sources have not previously been brought together in this fashion. The result of this work is new and novel insights, which will be of interest to researchers, practitioners, and policymakers.

The paper is presented as follows: first, we outline the published literature and explore the publicly available datasets to derive five hypotheses; next, we outline a methodological approach to analysing the data, and from here, we present our results, discussion, and conclusion along with calls for further research.

## 2. Theory and Hypothesis

The design of any curriculum is underpinned by human agency. It presents the students with what they should know and how they should come to know it [13–15]. There is notionally a gap between the planned curriculum, the delivered curriculum, and the experienced curriculum.

At a national level, the outcome of the planned EE curriculum in the UK is defined by the Quality Assurance Agency for Higher Education (QAA) as one which develops enterprising behaviours, attributes, and competencies leading to the creation of cultural, social, or economic value [16,17]. The QAA stresses that this does not have to focus solely on entrepreneurial outcomes, as the behaviours, attributes, and competencies are also likely to have a significant impact on the individual student in the context of their wider career.

With this in mind, we began to examine other literature spanning the planned, delivered and experienced curriculum to better understand what EEPs are, how they are structured, and what we already know about their impacts.

### 2.1. Entrepreneurship Education Programmes

In their discussion of the literature exploring entrepreneurship education in UK business schools, Matlay suggested that EEPs could be characterised by a focus on developing a substantial set of skills and knowledge required by entrepreneurs to support new venture creation over and above those required as an employee [18]. Further work broadened this significantly to include "any pedagogical program or process of education for entrepreneurial attitudes and skills, which involves developing personal qualities" [19]. This definitional difference is reflective of entrepreneurship education's position as a nexus business education discipline, which, although underpinned by knowledge and practice, is not dominated by a single approach [19].

This positioning of EE avails educators the freedom to shape EEPs into a variety of forms, from academic courses critically exploring entrepreneurship and its effects to highly applied venture creation programmes (VCPs) that are grounded in the practice of entrepreneurship and learning through this process [20]. Notably, EEPs are far fewer in number than other business education programmes [21], yet they present a complex field

of study [22], with only a relatively small number of papers exploring the structure of EEPs [23–25], their effectiveness [26–28], and their impact [7,9,18,29].

## 2.2. EEP Structure

A review of 205 entrepreneurship programmes in the USA [30] found that they had three main focuses: providing orientation and awareness, developing competencies for new venture creation, and exploring small business survival and growth. As we have already noted, more recent studies have shown that the field has changed radically in just under two decades, and there is now a range of programmes with differing aims and approaches for students to choose from [20]. That said, it is clear that the structure of these programmes and what this might mean for staff and students remains under-explored [23,24].

An exploration of 21 research articles on EEP content found that the most common subjects taught included: finance and marshalling of resources (16%), marketing (14%), idea development (13%), business planning (12%), management growth (12%), organisation and team building (10%), new venture creation (9%), SME management (8%), and risk and rationality (6%) [27]. It was also important to note the less common subjects, which were legal, management of innovation, negotiation skills, communication skills, and problem solving. A study [31] using a systematic review of 129 articles found that business planning, marketing, small business management, simulations, case studies, and networking were the most common components of an EEP. More recent work [32] analysing 50 MBA programs in entrepreneurship showed that the programs were primarily business and management, with a comparatively little focus on entrepreneurship itself.

A further review of empirical EE research claims that a set of interacting but not necessarily dependent clusters form the foundation of EE and, by extension, that these would be a useful model on which to base an EEP [33]. The clusters include entrepreneurial attributes, personality traits, learning, risks, motivations, and the theory of planned behaviour. However, to date, no empirical testing of this model has been performed.

Work by two of the authors of this paper reflected on their efforts to develop a new VCP in the UK [25]; therein, they proposed a 3-year undergraduate programme with a 50/50 split between academic study and practice. This structure, which may be argued reflects the journey of many early-stage entrepreneurs, transitions from a focus on mindset, ideation, and venture formation to operational management of an enterprise before finally challenging the students to consider growth, innovation, and raising investment.

A final study, ref. [23], draws on [34] alongside a comprehensive literature review of EEPs to develop a framework for their formulation. This comprised seven components, which were later refined in [24] with input from 38 international EE scholars; changes include a shift toward contextualisation instead of context, the recognition of the importance of entrepreneurship ecosystems as a component, and the development of the elements that underpin many of the components to reflect developments in practice. While this work offers a considered framing of EEPs, it lacks empirical testing and is very broad, presumably to reflect the range of approaches in the literature, which means that it may be of limited applicability in practice.

Taken together, these papers suggest a small but growing area of literature with several emergent models. Some of these share similarities in their objectives and/or approach, but it is clear that there is no one dominant norm beyond a grounding in business, which in some instances could be seen as the primary focus of the programmes.

All the existing models and studies lack longitudinal and empirical testing, and more problematically, much of this scholarship is based on secondary explorations of older programmes, which, in a rapidly developing field, may date quickly.

Our work herein does not seek to explain the reasons for this difference, but drawing on similar challenges already observed [33], we suggested that the differing forms are likely the result of teachers' (sic) diverse experiences and preferences shaping the programmes they deliver.

### 2.3. EEP Effectiveness and Impact

Several systematic literature reviews have been carried out to explore the impact of enterprise and entrepreneurship education [7,10,35–37]. These found generally positive relationships between engagement and entrepreneurship education or training and outcomes, such as entrepreneurial intent and business start-up. More recent research has suggested links between elective entrepreneurship education and the formation of entrepreneurial skills [38], the cultivation of entrepreneurial intent and self-efficacy via learning opportunities that lead to proof-of-concept and proof-of-business outcomes [39], and nascent entrepreneurial activity following engagement with simulations, workshops, and formal courses [40]. Other studies have found variable results, however, and suggest that entrepreneurship education may lead to a reduction in entrepreneurial intention [7], and where self-employment-related impacts have been found, they may be short-term or not sustained [41].

One study [18] found that research claiming that EEPs result in more nascent entrepreneurs after graduation was empirically unsupported and focused on small, biased samples measured only over the short term. More recently, a systematic literature review of EE [28] echoed these conclusions, noting that a specific focus on EE does not appear to guarantee effectiveness in the production of nascent entrepreneurs with increased entrepreneurial intention (EI).

Explorations of entrepreneurial intention have become more prevalent in the literature as researchers seek ways to investigate and articulate the impact of EEPs and other EE interventions. While these studies may have detractors who are critical of the approach on the grounds of its predictive ability [42], there is some recognition that, when applied longitudinally [43], these studies can provide a useful way of exploring how interventions might affect attitudes to action.

For example, [44] explored how three EEP components—learning, inspiration, and resources—can influence EI across 348 graduates from eight HEIs. They conclude that learning and inspiration activities can affect EI, helping students to perceive entrepreneurship as more relatable. However, they stressed that access to resources (incubation in this context) had the strongest impact on entrepreneurial intention.

A further study [27] found that any research of impact linked to the EEPs was inconsistent, although they noted three common success indicators (Others included technology transfer, employment creation, and student satisfaction):

1. Graduate start-ups;
2. Assessment scores;
3. Entrepreneurial mindset changes.

An additional review studied 32 papers and noted that EEPs containing entrepreneurial spirit and entrepreneurial skills appeared to correlate positively with the entrepreneurial ability of students [45]. However, their coding of entrepreneurial activity might be considered broad and only indicative at best.

Other research analysed Korean student satisfaction with a specific set of EEPs, which focus on the operational and financial aspects of business ownership [46]. This study highlighted the definitional challenge of EEPs and how that impacted student satisfaction, something noted previously by [26]. The authors also discussed setting programme objectives linked to an underpinning rationale, e.g., economic or social, which can then be measured to facilitate comparisons of impact, although this was not tested.

The work to date on impact and effectiveness in the context of EEPs suggests that while EI might be gaining traction as a methodological approach, the general discourse remains fractured. Furthermore, while these papers propose indicators such as start-ups, module results, mindset change [27], or student satisfaction [26,46], there is scope to develop these and space to establish baselines that could be compared more broadly to try to identify best practices.

From this research, we see that a focused, national study of EEPs has the potential to add significant value to the discourse, providing an exploration of the structure of EEPs

and measuring their collective effectiveness and impact. In the next section, we develop four hypotheses drawn from this literature, which we then use to do just that.

### 3. Hypothesis

#### 3.1. Programme Structure

As we have already established, EEPs are diversely designed with a range of elements to realise a variety of outcomes. That said, we know from some of the studies we have highlighted that, in any one programme, we might expect to find general business content, ancillary subjects (law, for example), and entrepreneurial content focused on ideation, venture design, formation, and development [5,27,32]. One would expect the focus of EEPs to be centred around teaching students about and/or through entrepreneurship, yet the only paper that quantifies this suggests a more balanced approach [27].

We are keen to understand how EEPs are structured in terms of the range of subject themes taught, as well as the number of entrepreneurship-related themes taught at each level within an entrepreneurship degree programme. Therefore, we hypothesise the following:

**H1:** *EEPs should be focused on entrepreneurship as their central subject theme.*

#### 3.2. Student Satisfaction

In our literature review, we noted that two studies [26,46] had used student satisfaction as a measure of the effectiveness of EEPs. Both highlighted the definitional challenges inherent in EEPs and noted that this could negatively impact student satisfaction. In broader studies, this sentiment was noted by several authors, including [47], who found that students who have well-defined expectations of the education they will receive are more satisfied with their programme.

A further study echoed this, stating that to manage the transition of students into university successfully, institutions need to be proactive in working to minimise any potential discrepancies between what students expect of university (and, by proxy, of their programme, lecturers, etc.) and what, in turn, is expected of them [48].

The suggestion from these papers is, therefore, that EEP students, studying programmes that present a range of pedagogy and demanding situations that they may never have experienced before, will encounter difficulties and, therefore, should be less satisfied [20]. We were interested to see if these observations translated across a broad cross-section of programmes, and if so, what might we learn from this about their effectiveness? Therefore, we hypothesis the following:

**H2:** *Entrepreneurship students should be less satisfied with their programmes than those on other business-related degrees.*

#### 3.3. Student Continuation

Work by [49] investigated the transition from a highly controlled, teacher-driven learning environment of schools or colleges to that of a university, noting that students' responsibility for their own learning is perhaps the biggest challenge they face. Their observations explore a new positionality of the student as a decision maker, choosing both their programme of study and the institution they will study at. The decision-making process has the potential to create mismatches between a student's expectations (generated through the programmes' marketing and engagement with its leaders) and the reality of the offer. Mismatches have the potential to colour the student experience and are likely to become evident in the first year of study, if not sooner.

This is because first-year experiences play a significant role in shaping students' attitudes and performance in subsequent years [50]. Several studies from Australia [51,52] have shown the importance of the first year and how this influences students' persistence

in higher education. This is why first-year dropout rates have become an important policy metric (e.g., [53]) and research area [54–56].

In other studies, Ref. [57] found that students in Columbia had lower dropout rates when the programme developed an improvement in students' entrepreneurial competitiveness. Ref. [58] suggested that entrepreneurship education is an appropriate "learning device" to reduce dropout rates at university, although their study only provides limited evidence for this.

This work suggests that first-year dropout rates may be an important metric through which the effectiveness of a programme can be measured. In England, this transition is referred to as continuation and means that one year and 15 days after first commencing their study [59,60], a student either:

- Continued—continued studies at the same higher education provider;
- Qualified—received a higher education qualification;
- Transferred—continued studies but at a different higher education provider.

Influenced by [57,58], we therefore hypothesise the following:

**H3:** *Entrepreneurship education programmes should have higher one-year continuation rates than other business-related degrees.*

### 3.4. Graduate Outcomes

Data on UK HEI-supported student and graduate self-employment and business start-ups are collected routinely through two key national surveys: the Higher Education Business and Community Interaction survey (HE-BCI) and the Graduate Outcomes (GO) cohort population survey. GO is completed by graduates 15 months after the completion of their programme. GO replaced the Destinations of Leavers of Higher Education (DHLE) survey completed 6 months after completion for the 2017/2018 graduating cohort. Institutions reported supporting 4058 graduate start-ups to the HE-BCI survey in 2020/2021, rising from 3892 in the previous year [61]. Data collection is problematic, however, and more complex and resourced measures may be needed [20,35]. Analyses conducted in preparation for this paper using HESA-provided data show that self-reported main activity for self-employment and business start-up is higher for GO than reported through HE-BCI, with 12,243 (3.2% of the cohort) of 2018/2019 graduate respondents giving their main activity as self-employed or freelance, and 8503 (2.2%) running their own business. A higher percentage (14.5%) reported that they are undertaking some element of self-employment, running a business, or portfolio development either full-time or part-time. Creative subjects show substantially higher levels of entrepreneurial activity than others. It is not known, however, if reported GO activity is a result of or influenced by students' educational experiences.

A new measure exploring graduate outcomes in the UK—Longitudinal Educational Outcomes (LEO)—was created in 2016 to explore graduate earnings one, three, or five years after graduation. An exploration of LEO focused on the effectiveness of the measure in understanding self-employment earnings showed that the percentage of graduates in self-employment increases with time after graduation [4.8% one year after graduation (2012/13 cohort); 5.8% three years after graduation (2010/11 cohort); and 6.7% 5 years after graduation (2008/09 cohort) [61]]. Data from graduates earning a salary from their own business through PAYE are not separated in LEO reports, and the self-employment rate alone is, therefore, an underestimate of graduate-owned business activity. Earnings through self-employment are substantially lower than those reported through PAYE [61]. Indeed, Universities UK has cautioned against the use of a LEO-driven funding model as institutions producing entrepreneurial graduates are not rewarded, and they have warned that such a model might restrict the growth of small businesses and start-ups in the arts and creative sectors [62].

Although the use of the LEO and similar salary information provided by graduates through GO is problematic when assessing employment-related outcomes for entrepreneur-

ship, it is used to provide information on the earning potential of graduates through UCAS and other public websites on the outcomes of programmes of study in the UK as an aide to help potential students assess which programmes to apply for. GO data is also used to provide insight into "paid employment" rates. Paid employment here includes those who are employed by others and those who are in self-employment [60]. Another term commonly used in relation to GO is "work", which includes paid, voluntary, and unpaid work [61]. Higher reported employment and work statistics based on GO might be expected for programmes that actively support entrepreneurship career options in addition to careers in an existing organisation as a result.

Therefore, we hypothesise the following:

**H4:** *Entrepreneurship education students will have higher rates of employability than students studying for other business-related degrees.*

**H5:** *Entrepreneurship education students will have lower initial earnings than students studying for other business-related degrees.*

### 4. Method/Methodology

The idea of benchmarking first appeared in business, where organisations sought to understand where they were in relation to their rivals. In recent years, the benchmarking of universities has become extremely common, from the generation of league tables [63] to the statistical comparison of clusters of universities to explore their research, teaching, and knowledge exchange activities [5,64]. This paper applies a benchmarking approach to explore the stated hypothesis not only because it is effective in comparative content such as the one we are working on but also because it reflects the process often applied by students and parents when judging the suitability of programmes.

In December 2021, the UCAS website was searched for undergraduate programmes that contained the word "entrepreneurship". The term "enterprise" was excluded from the search as it has several meanings that would have expanded the scope of the work beyond the stated frame.

The resulting programmes targeted students who had gained a Level 3 award and would be commencing further study in January or September 2022. The degree could be 3 or 4 years in length, and the awards could be BSc, BA, MA, BEng, or MEng. Top-up degrees were also recorded in the search. A total of 78 degrees were found, and the following data were collected for each programme of study. Item 1 was collected from the [65] website. Items 2 to 6 were collected from the programme information on universities' websites:

1. Programme Title/Name;
2. UCAS UG points required [65];
3. Programme Description;
4. Core Modules Names—Level 4 to 6. If there were Level 3 modules, these were not included, as the foundation year has a larger number of generic university-wide modules;
5. Teaching and Learning Statement;
6. Assessments Statement;
7. National Student Survey (NSS). (The UK National Student Survey (NSS, 2021), originally conceived to help potential students make informed choices, started in 2005. In the UK, it forms part of the Office for Students (OfS) quality assurance (QA) framework. The NSS is an anonymous, 27-question census of around half a million final-year HEI students mid-way through their final academic year. The published overall student satisfaction score was used in this analysis.) ([66]; NSS survey data);
8. Continuation Data as taken from the UK's Office of Student's [66];
9. DLHE-LEO Employability Data as taken from the UK's Office of Students DiscoverUni. (The Destination of Leavers from Higher Education (DLHE) collected information on a range of graduate outcome indicators six months after graduation. The Longitudinal Education Outcomes (LEO) dataset connects individuals' education data with their

employment, benefits, and earnings five years after graduation. Both are annual datasets across all UK universities. Since 2020, the DHLE has been replaced by the Graduate Outcomes Survey (GOS), data for which is collected fifteen months after graduation.) ([67]; 2017–2018 graduates);

10. Graduate Outcomes (GO) Data ([66]; from 2017/2018 graduating cohort).

The programme title was coded using the Higher Education Classification of Subjects (HECoS) [67], which has been implemented since 2019/20, and the Common Aggregation Hierarchy (CAH) [68], which provides a standardised hierarchical aggregation of HECoS codes based on three tiers of coding (levels 1, 2 and 3), allowing a clear structure for subject theme analysis.

HECoS has 1092 subject themes, and therefore CAH, with 26 high-level themes and 168 Level 3 themes, is an easier way to compare modules and programmes. However, CAH does not have a category for Enterprise or Entrepreneurship and maps this firstly to combined and general studies (CAH23) and then at a sub-level to Personal Development (CAH23-01-02), along with career guidance, research skills, and work-based learning.

The 78 programmes identified were categorised into the following CAH Tier 1 codes:

- (CAH03) biological and sport sciences = 2 programmes;
- (CAH07) physical sciences = 1 programme;
- (CAH10) engineering and technology = 3 programmes;
- (CAH11) computing = 1 programme;
- (CAH17) business and management = 56 programmes;
- (CAH23) combined and general studies = 9 programmes;
- (CAH25) design and creative and performing arts = 6 programmes.

This demonstrated that the majority (72%) of entrepreneurship undergraduate programmes still sat within the business management (CAH17) discipline.

Herein, the core, non-optional modules became the focus of the investigation in order to constrain the dataset and ensure we examine only those that are central to achieving the programme learning outcomes. Modules that were specific to non-business subject areas, such as microbiology or chemistry, were excluded. The resultant 1099 modules from the 78 Undergraduate programmes were categorised into one of the categories below, using Tier 3 of the CAH hierarchy to allow for a granular identification of the business-related subjects included within the programmes:

- Economics 15-02-01;
- Law 16-01-01;
- Business Studies 17-01-01;
- Marketing 17-01-02;
- Management Studies 17-01-03;
- Human Resource Management 17-01-05;
- Tourism, Transport, and Travel 17-01-06
- Finance 17-01-07;
- Accounting 17-01-08;
- Personal Development 23-01-02;
- Entrepreneurship 23-01-02 (HECoS Code 101221).

To categorise the modules within the CAH hierarchy, we developed the following approach as a principle: if a module name contains more than one subject term, the first subject term used in the title is taken as the principal theme of that module. Modules that contain entrepreneurship, venture creation, business planning, innovation, creativity, design thinking, business models, and raising capital were allocated to the entrepreneurship module grouping. To accommodate for the dilution effect highlighted by [69], modules that have the word "Entrepreneurial" or "Enterprising" attached to a term that might otherwise belong to another category (e.g., "Entrepreneurial Marketing") were allocated to the main subject theme ("Marketing" in this example). We understand that such modules may be more entrepreneurially focused, but an exploration at that level was beyond the scope of

this work. Finance (17-01-07) and Accounting (17-01-08) CAH groups were combined as a large number of modules contained both terms, such as Introduction to Accounting and Finance, Accounting and Finance Fundamentals, or Financial Accounting.

NSS score availability is contingent on having a sufficiently large number of respondents and on having been in place for a cohort of final-year students. Of the 56 EEPs with the Management Studies Group, NSS scores were available for 36 EEPs.

The analysis conducted firstly uses descriptive statistics, including the use of means, standard deviations, and percentages, to summarize and describe the features of the data (e.g., to describe the number of entrepreneurship modules in different programs, the average satisfaction scores, and the average earnings of graduates). T-tests were then used to determine if there were statistically significant differences between groups (e.g., in terms of student satisfaction or graduate earnings). To explore relationships between variables, such as the relationship between the number of entrepreneurship modules and student satisfaction, we used a Spearman correlation.

The paper also uses benchmarking to compare the EEPs against other programmes. This involves comparing metrics (like student satisfaction scores and continuation rates) against a standard or a set of best practices.

The statistical analyses reported below were conducted using SPSS Version 28.

## 5. Results

### 5.1. H1: EEPs Should Be Focused on Entrepreneurship as Their Central Subject Theme

To explore this hypothesis, we first examined the dataset created by profiling the core modules based on CAH subject themes across the 78 programmes in the sample. Table 1 analyses all EEPs, and Table 2 focuses specifically on those in the Management Studies Group.

The data in Table 1 shows that 29.5% (328) of core modules on all degrees in the sample focused on entrepreneurship, meaning that each programme contained, on average, a mean of 4.11 core entrepreneurship modules. The programmes in the sample had a mean of 14.26 core modules over the 3 years of the programme.

We then looked at degree programmes derived exclusively from the Management Studies Group, CAH17, to see if there was any difference (Table 2). These 56 programmes provided slightly more entrepreneurship content with, on average, a mean of 4.81 core entrepreneurship modules out of a programme mean of 15.93 core modules over the 3 years of the programme. (We were unable to account for the credit value of the modules in this analysis, as this was not consistently available across the dataset. We are aware that the weighting of modules by credit value may have revealed additional trends that we are not able to explore herein).

To further analyse this data, we looked at the number of core entrepreneurship modules at each academic level for the 58 programmes in the Management Studies Group (Table 3). Here, we observed that the first-year level (Level 4) of the degree contained the least number of core modules. The second year (Level 5) contained the largest number, but this was only slightly higher than the third year (Level 6).

**Table 1.** Subject themes within all EE programmes.

| | Entrepreneurship—23-01-02 | Business Studies—17-01 | Marketing—02 | Management Studies—03 | Human Resource Management—005 | Tourism, Transport and Travel—06 | Finance (07) and Accounting (08) | Law—16-01 | Economics—15- | Personal Development 23-01 | Other Modules |
|---|---|---|---|---|---|---|---|---|---|---|---|
| Module Count | 328 | 59 | 83 | 220 | 27 | 17 | 89 | 33 | 34 | 156 | 67 |
| Percentage | 29.5% | 5.3% | 7.5% | 19.8% | 2.4% | 1.5% | 8.0% | 3.0% | 3.1% | 14.0% | 6.0% |
| Mean | 4.11 | 0.79 | 1.06 | 2.64 | 0.35 | 0.21 | 0.69 | 0.44 | 0.39 | 0.46 | 1.90 |
| Std Dev | 2.66 | 0.93 | 1.02 | 1.99 | 0.56 | 1.06 | 0.82 | 0.63 | 0.70 | 0.75 | 1.78 |

**Table 2.** Subject themes within management studies EE programmes only.

| | Entrepreneurship—23-01-02 | Business Studies—17-01 | Marketing—02 | Management Studies—03 | Human Resource Management—005 | Tourism, Transport and Travel—06 | Finance (07) and Accounting (08) | Law—16-01 | Economics—15- | Personal Development 23-01 | Other Modules |
|---|---|---|---|---|---|---|---|---|---|---|---|
| Module Count | 279 | 57 | 75 | 195 | 25 | 16 | 83 | 30 | 32 | 132 | 12 |
| Percentage | 29.8% | 6.1% | 8.0% | 20.8% | 2.7% | 1.7% | 8.9% | 3.2% | 3.4% | 14.1% | 1.3% |
| Mean | 4.81 | 0.98 | 1.29 | 3.36 | 0.43 | 0.28 | 0.72 | 0.52 | 0.55 | 2.28 | 0.21 |
| Std Dev | 2.45 | 0.95 | 1.01 | 2.01 | 0.6 | 1.20 | 0.76 | 0.76 | 0.80 | 1.82 | 0.52 |

**Table 3.** Core entrepreneurship modules per academic level.

|  | Level 4 | Level 5 | Level 6 |
|---|---|---|---|
| Sum. of entrepreneurship modules | 53 | 119 | 107 |
| Mean no. of entrepreneurship modules per programme | 0.91 | 2.05 | 1.84 |
| SD entrepreneurship module | 1.19 | 1.36 | 1.21 |
| Min | 0.00 | 0.00 | 0.00 |
| Max | 5.00 | 6.00 | 7.00 |

This pattern would seem to suggest a move toward specialisation in the programmes as they develop across the levels. The dip at Level 6 is possibly the result of optional modules, which may be more common at this level of study, and/or the dissertation/project/extended essay module, which is likely to be related to the specialism of the programme, but may not have an "entrepreneurship" marker.

As described above in the Methodology, 36 out of the 58 Management Studies Group programmes had been in place long enough and had sufficient respondents for their NSS student satisfaction scores to be published online. Student satisfaction scores were not available for 22 programmes. It may be argued that some of the programmes dedicated to entrepreneurship, such as Venture Creation Programmes [20], are most likely to be made up of small cohorts and may not have an associated student satisfaction score as a result. To explore this possibility, the number of entrepreneurship modules for programmes with and without student satisfaction scores is given in Table 4. Here, we can see that the mean number of modules for programmes without an available student satisfaction score is indeed higher at each level than for those with a score.

**Table 4.** Core entrepreneurship modules with and without NSS student satisfaction per academic level.

|  | With Student Satisfaction | | | Without Student Satisfaction | | |
|---|---|---|---|---|---|---|
|  | Level 4 | Level 5 | Level 6 | Level 4 | Level 5 | Level 6 |
| No. of Entrepreneurship Modules | 27 | 72 | 62 | 26 | 47 | 45 |
| Mean | 0.8 | 2.0 | 1.7 | 1.2 | 2.1 | 2.0 |
| SD | 1.2 | 1.3 | 1.0 | 1.2 | 1.5 | 1.5 |
| Min | 0 | 0 | 0 | 0 | 0 | 0 |
| Max | 5 | 6 | 4 | 3 | 6 | 7 |

Taken together, the tables suggest that entrepreneurship content is likely the largest core subject theme, although the standard deviation does indicate some variation across the programmes. Although the data shows that each programme contains a relatively small number of core entrepreneurship modules in comparison to the total number of core modules of which the programme is composed, we can observe a trend toward increasing numbers of core entrepreneurship modules as the programmes develop.

The absolute number of entrepreneurship modules within a programme gives a less than conclusive position with respect to H1. It could be argued that entrepreneurship is the largest subject theme, so it should be considered the focus of a program. However, it is also clear that entrepreneurship modules do not necessarily represent the majority of *core* content delivered on the programmes considered here. Perhaps this pattern is unique to entrepreneurship as a focus, reflective of its location at a nexus of ideas, approaches, and practices in business. Alternatively, it may be similar to other degrees in different disciplines with a similar subject focus, such as marketing. Without comparative data as a baseline across other programmes or a deeper credit/content analysis, any claims as to

focus are hard to definitively substantiate from our data, meaning that H1 is neither proved nor disproved.

### 5.2. H2: Entrepreneurship Students Should Be Less Satisfied with Their Programmes than Those on Other Business-Related Degrees

Overall, NSS student satisfaction across the English HEI sector was 82.15% across all programmes in 2020 [70]. For business degrees, this was 81% [71]. The 38 EEPs analysed, for which a reported student satisfaction score from the NSS results is available, had a mean overall satisfaction score of 74.1%, a significant drop of 8.1% compared to the national average [$t(35) = -6.09$; $p < 0.01$], and 6.9% for business degrees [$t(35) = -5.30$; $p < 0.01$].

To develop this further and explore if there was a relationship between student satisfaction and the inclusion of the entrepreneurship modules, the overall student satisfaction score was Spearman correlated with the percentage of entrepreneurship modules. It was found that there was no statistically significant relationship between student satisfaction and the percentage of entrepreneurship modules across the whole programme.

Data from the NSS shows that EEP students are less satisfied than their counterparts studying on other business degrees and the wider university sector as a whole; as such, H2 is proven. However, as discussed above, EEPs with available student satisfaction scores have fewer entrepreneurship-specific modules than those without; programmes with the most entrepreneurship-specific content are therefore excluded from this analysis, which may skew the results.

### 5.3. H3: Entrepreneurship Education Programmes Will Have Higher One-Year Continuation Rates than Other Business-Related Degrees

The average one-year continuation rate for EEPs is 85.1%. This means that EEPs lose 14.9% of their students by the end of the first year of study. To put this in perspective, across all UK universities and all subjects, 6.7% (93.3%) of UK full-time first-degree entrants in 2018/19 did not continue past their first year, and for all Business and Management degrees, this rate is slightly lower at 6.6%. This demonstrates that hypothesis H3 is not proven. EEPs have significantly lower continuation rates, which are below both the national average [$t(36) = -8.07$; $p < 0.01$] and the average for business and management degrees [$t(36) = -8.16$; $p < 0.01$].

In an effort to explore what else the data might reveal as to why this is, we first looked to see if there was any association between one-year completion rates and the amount of entrepreneurship in the curriculum. The analysis showed that there was no relationship between the one-year completion rate and the percentage [$r(37) = 0.032$, $p = ns$] of first-year entrepreneurship modules, indicating that the amount of entrepreneurship taught does not appear to be a factor.

Prior research has demonstrated a general relationship between UCAS entry points obtained to study a programme and completion rates: the higher the qualifications for entry, the higher the completion rates [72]. The data here shows a similar trend. Programmes requiring higher UCAS entry points are more likely to have higher continuation rates [$r(35) = 0.35$, $p < 0.05$]. The data provided also enables us to break down the source of these UCAS points into A Levels, Baccalaureate, Foundation Year, or a different higher education qualification. For this year of analysis, not all programmes accepted all or reported all qualifications. These correlations show that A Level based entry points [$r(36) = 0.15$, $p = ns$] and points obtained from a different higher education qualification [$r(36) = -0.26$, $p = ns$] have no relation to first-year completions. Foundation Year programme entry points positively correlate to higher continuation rates [$r(35) = 0.38$, $p < 0.05$], as do Baccalaureate entry [$r(35) = 0.35$, $p < 0.05$].

This shows that EEPs with higher entry requirements have higher one-year completion rates, although these are still below the national average. It is clear that the amount of first-year entrepreneurship content in the programme is not a factor in continuation, but those students with prior experience of programmes structured in a degree-level format,

the group who studied foundation degrees herein appear more likely to continue than those with other qualifications on entry.

### 5.4. H4: Entrepreneurship Education Students Will Have Higher Rates of Employability than Students Studying for Other Business-Related Degrees

Based on the GO survey [73], employment for EEPs is 89.0% at 15 months post-completion. This rate is significantly higher than the overall rate for all other business degrees, where the mean is 86.30% [t(27) = 2.19; $p < 0.05$]. Although the EEP employability rate is lower than the mean rate across all UK undergraduate degrees of 90.7%, the difference does not reach significance [t(27) = −1.11; $p$ = ns]. This supports H4, as EEP students have higher rates of employability than those who are studying for other business-related degrees. Although the hypothesis is supported, the data is not indicative of a significant effect on comparative rates of employability overall, as these lag behind the national average for all graduates.

### 5.5. H5: Entrepreneurship Education Students Will Have Lower Initial Earnings than Students Studying for Other Business Management Degrees

HESA's GO data report GBP 24,979 as being the average salary 15 months after graduation for all graduates in all subjects. The average salary reported for the sample of EEP explored here is indeed significantly lower at GBP 23,397 [t(28) = −4.17; $p < 0.01$].

According to the Chartered Association of Business Schools analysis of the 2018/19 GO survey of UK-domiciled first-degree graduates in Business and Administrative Studies [71], 28% earned less than GBP 20,999, 40% reported a salary of between GBP 21,000 and GBP 29,999, an additional 11% recorded a salary of between GBP 30,000 and GBP 38,999 (marginally behind the average of 13% for all graduates across all subjects), and 3% reported a salary of GBP 39,000 or higher.

If, as [71] suggests, the pattern for all Business and Administrative Studies graduates matches the average results for first-degree graduates across all subjects, then we can infer that the average salary for EEP students reported above is also significantly lower than for students studying on business and management degrees. In that case, then H5 is proven, although the reasoning and mechanism behind this remain unclear.

## 6. Discussion

This study is the first of its kind, certainly in a UK context. It uses a range of data sources and makes significant strides in understanding EEPs, offering a unique analysis of their structure, effectiveness, and impact on student outcomes.

Our findings reveal that while the EEPs we sampled were predominantly focused on entrepreneurship, they encompassed a relatively small proportion of entrepreneurship-specific modules. This insight supports prior conclusions [32] and further challenges the perception of EEPs [23,24], suggesting that additional work to explore curricula is needed if these programmes are to truly be seen as occupying their own niche in the wider business discipline.

That said, without comparative data exploring other discipline-specific degrees (e.g., marketing), it is difficult to understand if this result is unique to EEPs or not. Therefore, benchmarking exercises comparing EEP structure with other explicitly named disciplines, such as marketing programmes, would be beneficial.

Notably, our research highlights a trend toward increasing entrepreneurship content as students progress through their programmes, albeit with a dip in the final year. This progression suggests a gradual immersion into entrepreneurship, which could be designed to shape student competencies and career aspirations. That said, it could also be a feature of educational design in a UK context, which, as noted, we cannot establish. Certainly, this appears to be an accepted principle that educators are applying as the basis of their work in developing programmes.

Frustratingly, we note several challenges in this specific data, from the exclusion of some programmes in the dataset due to its collection parameters to the possibility that

trends over time may be skewed by module naming conventions. For example, a final year dissertation can be entrepreneurship-specific but not identifiable from the module name, meaning that the EEP-related topics identified by [27,31] may also be masked.

A critical finding of our study is the lower student satisfaction in EEPs compared to other business-related degrees. This could be indicative of the unique challenges and innovative pedagogies inherent in entrepreneurship education and evidence of entrepreneurial learning [20], which students may not fully grasp or for which they may not have been properly prepared; we can only speculate herein. What this underscores is the importance of aligning student expectations with the realities of entrepreneurial education, suggesting a potential area for improvement in student engagement and support. To our knowledge, this represents the first instance of these phenomena being analysed and discussed using a national dataset.

Contrary to expectations, our study found that EEPs do not demonstrate higher first-year continuation rates. This finding is significant as it suggests that the allure of entrepreneurship alone may not be sufficient to retain students or that they are underprepared for the realities of the challenges faced in an EEP context. It certainly highlights the need for a deeper exploration into the factors influencing student retention in EEPs, such as entry requirements and the academic background of students. As our findings run contrary to the limited literature suggesting that EEPs can reduce dropout rates [57,58], this is a very important point that merits further investigation.

From an employability perspective, EEP graduates show promising trends with higher employability rates compared to their counterparts in other business degrees. However, they also face lower initial earnings. We suggest that they are more "employed" because many are likely to be working for themselves, but they earn less initially doing so as they seek to develop their ventures. This is likely a reflection of an entrepreneurial career path that involves building businesses from the ground up. This finding is crucial for prospective students and educators, as it sets realistic expectations regarding the entrepreneurial journey post-graduation. It is also important to note that whether income levels improve over time is impossible to say as longitudinal data across the sector does not exist, which in and of itself should be a challenge for future research.

## 7. Conclusions

This paper has taken a novel approach in combining a range of datasets to present a unique perspective on the structure and impacts of EEPs in the UK. This research contributes significantly to the discourse on entrepreneurship education, providing new insights and raising important questions about EEPs, which should act as provocations for further research.

We would suggest that the most important aspect of this paper is the under-researched trends that it highlights in both the structural assessment and the impact measurement of EEPs, along with the methods we use to explore these.

When focusing on EEPs, our work presents several uncomfortable results that educators, policymakers, and institutions should consider; specifically, those designing EEPs may need to reconsider their structure, and current received practice may not be sufficient to meaningfully impact student outcomes. It is quite apparent from the data that students are not satisfied or perhaps not prepared correctly for their experience; this leads to higher levels of disengagement, and this is not being addressed nationally in a way that affects the data we examined. This may also call into question the suitability of the data collection measures. Furthermore, the reward for those who do stay the course in an EEP appears less enticing in the short term, which may raise questions about EEPs more generally as vehicles on which to build graduate-level careers.

We encourage the enterprise and entrepreneurship education community to consider these findings in the ongoing development and refinement of EEPs. We believe that by doing so, they could further explore the design of their courses and the metrics used to capture impact. Together, these might better prepare students for the challenges and

opportunities of entrepreneurial careers, ultimately contributing to the broader landscape of innovation and economic growth.

Through this work, we identified a range of calls for potential further research; these include the following:

- Quantitative studies that conduct similar benchmarking exercises in other disciplines to allow for comparative analyses between programmes.
- Quantitative and qualitative studies that explore EEP curriculum design in more depth to build on the trends noted herein, especially around pedagogy, practice, and the balance of modules.
- Qualitative studies on a national and subject level that focus specifically on whether students are adequately prepared for their course and the impact this might have.
- Quantitative and qualitative studies that explore the impact of EEPs longitudinally to see if longer term career and earning prospects are greater for groups studying these pathways.

Our study's limitations include the scope of the data analysed and the generalisability of our findings, which we note throughout. Despite these limitations, the paper brings together previously understudied datasets to draw new conclusions. We feel that the understanding we have reached has value and opens a range of avenues for future research.

**Author Contributions:** Conceptualization, M.R.-D. and D.B.; methodology, M.R.-D., D.B. and K.S.; software, D.B.; validation, D.B. and K.S.; formal analysis, M.R.-D., D.B. and K.S.; investigation, M.R.-D., D.B. and K.S.; resources, D.B., K.S. and M.M.; data curation, D.B. and K.S.; writing—original draft preparation, M.R.-D., D.B., K.S., M.M., V.C. and D.M.; writing—review and editing, M.R.-D., D.B. and K.S.; visualization, D.B.; supervision, M.R.-D.; project administration, M.R.-D. All authors have read and agreed to the published version of the manuscript.

**Funding:** This research received no external funding.

**Institutional Review Board Statement:** Not applicable.

**Informed Consent Statement:** Informed consent was not required as this article uses open-source data.

**Data Availability Statement:** All data used in this article is openly available through the citations.

**Conflicts of Interest:** The authors declare no conflicts of interest.

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
