# Peer review of "Does Entrepreneurship Education Deliver? A Review of Entrepreneurship Education University Programmes in the UK"

_education, doi:10.3390/educsci14040361_

Round 1

Reviewer 1 Report

Comments and Suggestions for Authors

This paper is welcomed and is well written.  The hypotheses are clear and concise.  The methodology is cleverly designed.  The review of the relevant literature is thorough, valid and reliable.  The overall finding are informative, clear and provide significant opportunity to reflect on them and to consider them further.  Any comments for improvement would centre on review of more recent literature to be included.  The conclusions and recommendations require work and can be expanded and explored much more than presented in the paper.  It is as if the author(s) became weary and lacked enthusiasm to deliberate more and contribute more accordingly.  It is recommended that this be revisited to do justice to the research conducted and to generate a significant number of topics for future research, discussion and debate and most importantly to add value to the impact of entrepreneurship education.  Suggestions for targeted empirical research at both a quantitative and qualitative nature would be very welcome.  Looking forward to its publication, discussions generated and future research.

Author Response

Please find the response in the attached document. Thank you!

Reviewer 2 Report

Comments and Suggestions for Authors

Dear Authors,

The paper deals with an interesting and up-to-date topic and contains useful information. However, I have some suggestions related to the consistency and clarity of the paper:

  1. Within the text you use the terms program, course, module, and degree yet sometimes the meanings overlap or are used ambiguously. Please try to make it clearer. E.g. line 355 you mention courses, then in line 408 you refer to them as programs and in line 334 you even mention them as degrees. This makes it confusing as the terms have different meanings.
  2. In line 385 you said CAH23-01 is personal development yet CAH23-01 is combined and general studies.  In line 386 you said CAH23-01-02 is Entrepreneurship yet CAH23-01-02 is personal development. For me, it is confusing how you did this classification and why you classified Entrepreneurship under CAH23-01-02. Entrepreneurship truly involves personal development to some extent, but in my opinion, it primarily falls within the realm of business and management studies rather than personal development, as it involves starting, managing, and growing a business.
  3. In line 395 you said that you allocated “Entrepreneurial Marketing” to the marketing theme, yet I think that to classify the course correctly you need to analyze the course curriculum, as it can be either part of marketing or entrepreneurship.
  4. In line 400 you say “out of the 58 EEPs”. Where is this number coming from? It does not show up in the text how you reached that number.  
  5. I feel that you failed to mention the statistical methods you used to support your hypothesis, and how they specifically apply to the research questions.
  6. Table 2, line 424, if the table refers only to the CAH17 subjects why are the other study themes included in the table? Also, I believe that a more detailed explanation accompanying each data representation will enhance clarity.
  7. It would be useful to explain what you mean in H1 when you say “Entrepreneurship courses should be focused on teaching entrepreneurship”? Do you refer to entrepreneurship courses or entrepreneurship programs and how do you define teaching entrepreneurship? Also, I do not feel that the way you approached it does not help in proving or disproving your hypothesis. A more in-depth analysis is needed here.
  8. H2 analyzed as such I think is very subjective. It would be also useful to analyze the courses taught at the EEPs that you analyzed and the business-related programs, as you mentioned that the EEPs, according to the literature, also contain general business content and ancillary subjects. What is the difference in the curricula? It would help to argue your hypothesis better.
  9. In the Conclusion chapter you acknowledge some limitations, but a more comprehensive discussion of the limitations and their impact on the findings would add to the paper's credibility.

Best regards,

Author Response

(The authors gave the same response as above.)

Reviewer 3 Report

Comments and Suggestions for Authors

Please revise the following sections:

- The methodology needs further explanations about content analysis and statistical analysis.

- What is the contribution of the research for educational designers or policymakers?

- The conclusion and discussion sections need to be rewritten

Author Response

(The authors gave the same response as above.)

Reviewer 4 Report

Comments and Suggestions for Authors

The paper is not well-structured and it is difficult to understand the mean of the research, which is confusing. Moreover, it is rather descriptive.

Author Response

  • I would contend that the paper has a clear structure which three other reviewers have recognised.
  • The paper itself may be difficult to follow in the opinion of this reviewer, but this appears to be a minority view. Any comments in this vein from other reviewers have been actioned.
  • The paper has a clear analytical approach and structure which develops a series of novel conclusions that other reviewers have recognised.

 Thank you! 

Round 2

Reviewer 3 Report

Comments and Suggestions for Authors

The revisions are done correctly.

Author Response

Thank you for your suggestions and help.

Reviewer 4 Report

Comments and Suggestions for Authors

The paper has improved after reviewers' comments. Go ahead with publication.

Author Response

(The authors gave the same response as above.)
